



**Impacts of Stratospheric Sulfate Geoengineering on Tropospheric Ozone**
Lili Xia[1*], Peer J. Nowack[2], Simone Tilmes[3], and Alan Robock[1]
[1]Department of Environmental Sciences, Rutgers University, New Brunswick, New Jersey, USA
[2]Centre for Atmospheric Science, Department of Chemistry, University of Cambridge,
Cambridge, UK
[3]Atmospheric Chemistry Division, National Center of Atmospheric Research, Boulder, CO, USA

*To whom correspondence should be addressed: Lili Xia, Department of Environmental Sciences,
Rutgers University, 14 College Farm Road, New Brunswick, NJ 08901-8551. E-mail:
lxia@envsci.rutgers.edu.


24                                  **Abstract**

A range of solar radiation management (SRM) techniques has been proposed to counter

anthropogenic climate change.  Here, we examine the potential effects of stratospheric sulfate
aerosol and solar insolation reduction on tropospheric ozone and ozone at Earth's surface.
Ozone is a key air pollutant, which can produce respiratory diseases and crop damage.  Using a
version of the Community Earth System Model from the National Center for Atmospheric
Research that includes comprehensive tropospheric and stratospheric chemistry, we model both
stratospheric sulfur injection and solar irradiance reduction schemes, with the aim of achieving
equal levels of surface cooling relative to the Representative Concentration Pathway 6.0 scenario.
This allows us to compare the impacts of sulfate aerosol and solar dimming on atmospheric
ozone concentrations.  Despite nearly identical global mean surface temperatures for the two
SRM approaches, solar insolation reduction increases global average surface ozone
concentrations while sulfate injection decreases it.  A key difference between the two
geoengineering schemes is the importance of heterogeneous reactions in the photochemical
ozone balance with larger stratospheric sulfate abundance, resulting in increased ozone depletion
in mid- and high latitudes.  This reduces the net transport of stratospheric ozone into the
troposphere and thus is a key driver of the overall decrease in surface ozone.  At the same time,
the change in stratospheric ozone alters the tropospheric photochemical environment due to
enhanced ultraviolet radiation.  A shared factor among both SRM scenarios is decreased
chemical ozone loss due to reduced tropospheric humidity.  Under insolation reduction, this is
the dominant factor giving rise to the global surface ozone increase.  Regionally, both surface
ozone increases and decreases are found for both scenarios, that is, SRM would affect regions of
the world differently in terms of air pollution.  In conclusion, surface ozone and tropospheric
chemistry would likely be affected by SRM, but the overall effect is strongly dependent on the



SRM scheme. Due to the health and economic impacts of surface ozone, all these impacts
should be taken into account in evaluations of possible consequences of SRM.



# 1 Introduction

## 1.1 Atmospheric Ozone

It is well known that sulfate aerosols in the stratosphere enhance heterogeneous chemical reactions that lead to ozone depletion. With present day anthropogenic halogen loading, the aerosols provide surface area for heterogeneous reactions that activate halogens and hence increase catalytic ozone destruction, especially in high latitudes (Tie and Brasseur, 1995). This has been modeled and observed following the large 1982 El Chichón and 1991 Pinatubo volcanic eruptions (Tie and Brasseur, 1995; Portman et al., 1996; Solomon, 1999).

However, volcanic eruptions do not only affect stratospheric ozone, but also impact tropospheric composition, often due to stratosphere-troposphere coupled effects. The 1991 Pinatubo eruption, for example, has been linked to changes in stratosphere-troposphere exchange (STE) of ozone (Aquila et al., 2012; Aquila et al., 2013; Pitari et al., 2016). In addition, the stratospheric ozone decrease led to an invigorated photochemical environment in the troposphere due to enhanced downward chemically-active ultraviolet (UV) radiation (Tang et al., 2013).

This study focuses on tropospheric ozone, in particular surface ozone concentration changes. Surface ozone is of central importance to Earth's environment and as an air pollutant it adversely impacts human health (e.g., Kampa and Castanas, 2008) and the ecosystem (e.g., Mauzeral and Wang, 2001; Ashmore, 2005; Ainsworth et al., 2012). There have been numerous studies of the observed surface ozone trend (e.g., Cooper et al., 2014), identifying ozone sources and sinks (e.g., Wild, 2007), predicting future changes (e.g., Young et al., 2013), and understanding the impacts of such changes (e.g., Silva et al., 2013). Global surface ozone concentrations are estimated to have doubled since the preindustrial period (Vingarzan, 2004), mainly due to increased emissions of ozone precursors associated with industrialization (e.g.,



Forster et al., 2007). Differences in future tropospheric ozone concentrations will be strongly
dependent on the emission pathway followed (Stevenson et al., 2006), which will determine both
in-situ tropospheric chemical production of ozone and transport from the ozone-rich stratosphere
(Collins et al., 2003; Wild et al., 2012; Neu et al., 2014).
**1.2 Differences between sulfate and solar geoengineering**

The progression of global warming, slow mitigation efforts, and our relatively limited

adaptive capacity, force consideration of SRM geoengineering as one possible strategy to avoid
many of the most undesirable consequences of global warming (Crutzen, 2006; Wigley, 2006;
Tilmes, 2016a). The above discussed factors controlling tropospheric ozone concentrations
could be affected by SRM schemes (Nowack et al., 2016). Here we compare a proposed
geoengineering scheme, stratospheric sulfur injection, to solar irradiance reduction. Both
schemes would cool Earth's surface by reducing sunlight reaching the surface, either by aerosols
reflecting sunlight or by artificially reducing the solar constant in a climate model, but sulfate
geoengineering would strongly heat the stratosphere and provide aerosol surfaces for chemical
reactions. Previous studies have shown that injected sulfur chemically forms sulfate aerosols
within a couple of weeks. The aerosol layer absorbs near infrared solar radiation as well as
outgoing longwave radiation and results in stratospheric warming (e.g., Tilmes et al., 2009;
Ammann et al., 2010; Jones et al., 2011). Additionally changes in ozone and advection impact
the warming in the stratosphere (Richter et al., 2017, submitted). Under solar reduction, the
stratosphere would be cooler due to reduced shortwave heating (Govindasamy and Caldeira,
2000), although simultaneous stratospheric ozone changes (if considered) may buffer this effect
(Nowack et al., 2016).



One of the most important differences between the two scenarios is that if a permanently
enhanced stratospheric aerosol layer is artificially created in an attempt to reduce anthropogenic
global warming, the resulting strong ozone depletion, in particular in mid- and high latitudes,
would have serious impacts on the biosphere, similar to the effects observed after large volcanic
eruptions described above (Crutzen, 2006; Rasch et al., 2008a; Rasch et al., 2008b; Tilmes et al.,
2008, 2009, 2012).  This effect would have to be expected as long as there is anthropogenic
halogen in the stratosphere.  In the remote future, the decreasing burden of anthropogenic
halogen components will eventually result in an increase in stratospheric ozone due to the
importance of heterogeneous reactions to the nitrogen cycle in the upper stratosphere, which
increases ozone in the middle and upper stratosphere (Tie and Brasseur, 1995; Pitari et al, 2014).
Overall such changes to the stratosphere would also have important implications for tropospheric
composition.  Decreasing stratospheric ozone leads to more UV propagating through, with
increasing ozone having the opposite effect, which would thus alter the photochemical
environment of the troposphere in different ways (Tilmes et al., 2012; Nowack et al., 2016).
In the following sections, we describe the experimental set-up of the two geoengineering
schemes and discuss some general climate impacts, followed by a detailed discussion of
tropospheric and surface ozone changes.  We also show that sulfate and solar geoengineering
would impact the stratosphere differently, which implies further key differences in their potential
influences on tropospheric composition.  In this study, we examine the impacts of stratospheric
sulfate geoengineering on tropospheric ozone for the first time.
**2  Model and Experiment Design**
We simulated both types of SRM schemes using the full tropospheric and stratospheric
chemistry version of the Community Earth System Model – Community Atmospheric Model 4




(CESM CAM4-chem) with horizontal resolution of 0.9° x 1.25° lat-lon and 26 levels from the
surface to about 40 km (3.5 mb).  The model has been shown to give a good representation of
present-day atmospheric composition in the troposphere (Tilmes et al., 2016b) and stratosphere
at 2° resolution (Fernandez et al., 2017).  Similar to the 2° model version, the 1° horizontal
resolution version of the model also produces reasonable stratosphere and troposphere ozone
chemistry (Figs. S1-S2).  CAM4-chem is fully coupled to the Community Land Model version
4.0 with prescribed satellite phenology (CLM4SP), the Parallel Ocean Program version 2 (POP2)
ocean model and the Los Alamos sea ice model (CICE version 4).  The model uses a nudged
quasi-biennial oscillation (QBO), which means the QBO will not be modified by the radiative
interaction of the aerosol.  Interaction between aerosol burden and photolysis rates is not
included in the model.  Changes in photolysis rates in the troposphere are calculated depending
on the total ozone column change (Kinnison et al., 2007).  Volatile organic compound (VOC)
emissions are simulated by the Model of Emission of Gases and Aerosols from Nature (MEGAN
v2.1) (Guenther et al., 2012).  The dynamical ocean model does not include any biogeochemical
feedbacks and only the atmospheric and land models are coupled to the atmospheric chemistry
component.  The model configuration used here, at 2° resolution, is participating in the current
phase of the Chemistry-Climate Model Initiative (Tilmes et al., 2016b).

We compare three ensemble members each of the two geoengineering scenarios to a

three-ensemble reference run with Representative Concentration Pathway 6.0 (RCP6.0;
Meinshausen et al., 2011) anthropogenic forcing from 2020 to 2089.  Our sulfate aerosol
implementation is the G4 Specified Stratospheric Aerosol (G4SSA) experiment (Tilmes et al.,
2015), whereas solar reduction geoengineering is the solar analog (hereafter G4SSA-S) by
imposing a solar irradiance reduction with the same negative radiative forcing at the top of the





142 atmosphere (TOA) as in G4SSA. G4SSA uses a prescribed stratospheric aerosol surface area

143 distribution to mimic the effects of continuous emission into the tropical stratosphere at 60 mb of

144 8 Tg $SO_2$ $yr^{-1}$ from 2020 to 2069. More details of this prescribed stratospheric aerosol are given

145 in Tilmes et al. (2015b) and Xia et al. (2016). The G4SSA scenario then continues from 2070 to

146 2089 without imposed aerosols to study the termination effect of geoengineering. During the

147 sulfate injection period, the net solar flux at the TOA has been decreased by 2.5 $W/m^2$ compared

148 to RCP6.0 (Fig. 1a). This number was obtained by a double radiation call in the model in

149 calculating the direct forcing of the prescribed aerosol layer. To attain the same TOA solar flux

150 reduction in G4SSA-S, we reduced the total solar insolation by 14.7 $W/m^2$ during 2020-2069

151 assuming a global average planetary albedo of 0.32 ($14.7\ W/m^2 = \frac{2.5\ W/m^2 \times 4}{1.0 - 0.32}$) (Fig. 1b). From

152 2070 on, we accordingly reset the total solar insolation back to the reference level to simulate the

153 abrupt termination of geoengineering.

154 **3  Results and Discussion**

155 **3.1 Climatology in G4SSA and G4SSA-S**

156  As a consequence of the same net all-sky TOA solar flux reduction in G4SSA and

157 G4SSA-S (Fig. 1a), the two scenarios show approximately the same global mean surface

158 temperature reduction of 0.8 K compared with RCP6.0 (Fig. 2a) (all values below are the

159 average of years 2030-2069). After the termination of geoengineering on 1 January 2070, the

160 global mean surface temperature rapidly increases. Fig. 3 shows the surface temperature

161 differences between G4SSA, G4SSA-S, and RCP6.0 in years 2030-2069 (the last 40 years of

162 geoengineering). Consistent with the global average temperature change, the two

163 geoengineering scenarios have similar temperature reduction patterns (Fig. 3a and 3b), and the

164 differences between them are not significant in most regions (Fig. 3c). The warming in northern



Europe and Asia shown in Fig. 3c is stronger in Northern Hemisphere winter (Fig. S3), which is
the characteristic "winter warming" from volcanic stratospheric aerosol (Robock, 2000).
However, the zonal mean stratospheric temperatures in G4SSA and G4SSA-S differ substantially
(Fig. 4).  As shown in previous studies (Tilmes et al., 2009; Ammann et al., 2010; Jones et al.,
2011), sulfate aerosol in the stratosphere results in strong warming by 3 K in the tropics (Fig. 4a),
while in G4SSA-S there is slight cooling (Fig. 4b).  In both cases, the troposphere shows strong
temperature reduction with similar patterns and ranges.

Global averaged precipitation and evaporation have similar size reductions of 0.07

mm/day in the two scenarios (Fig. 2c and Fig. S4), with no statistically significant difference
between them.  Most of the evaporation terms show a larger reduction in G4SSA than in
G4SSA-S, except for plant transpiration, which has the opposite pattern (Fig. S4).  As shown by
Xia et al. (2016), enhanced diffuse radiation in G4SSA increases photosynthesis, which produces
stronger transpiration.  Therefore, transpiration in G4SSA reduces less than in G4SSA-S.

The similar evaporation reduction in G4SSA and G4SSA-S can also be explained by the

surface energy budget (Fig. 5).  Although we keep the net shortwave radiation at the TOA the
same in the two schemes (Fig. 1a), surface net solar radiation reduces more in G4SSA than in
G4SSA-S (Fig. 2b and Fig. 5) due to the absorption by sulfate aerosol in the near-infrared.  This
stronger surface solar forcing in G4SSA-S is mainly balanced by larger net longwave radiation to
the atmosphere (Fig. 5).  As a result, latent heat changes in the two scenarios are similar.

The precipitation and evaporation changes in this study are different from previous

studies by Niemeier et al. (2013) and Ferraro et al. (2014).  In their studies, with the same
magnitude of surface cooling, the sulfate injection scenario led to a greater reduction of globally
averaged evaporation and precipitation as compared with the solar case.  They attribute this





result to extra downwelling longwave radiation because of stratospheric heating from the
injected aerosol, which would heat the upper troposphere and stabilize the troposphere compared
with the solar reduction case (which we did not find, Fig. 4c), and result in a stronger reduction
of latent heat and precipitation in sulfate injection geoengineering.  We find two possible reasons
for the differences: (1) The column ozone change plays an important role.  In Niemeier et al.
(2013) and Ferraro et al. (2014), the same prescribed ozone was used in all scenarios, while we
used a fully coupled atmosphere-chemistry model.  As shown in section 3.2, total column ozone
in G4SSA reduces by about 5 DU (mainly in the lower stratosphere) compared with RCP6.0 and
G4SSA-S (Fig. 6).  Less ozone in G4SSA will change its radiative forcing, surface radiative
fluxes and atmospheric lapse rate (Chiodo and Polvani, 2015; MacIntosh et al., 2016; Nowack et
al., 2015, 2017) and thus contribute to the differences between the two studies. (2) Enhanced
transpiration in G4SSA due to enhanced diffuse radiation reduces the evaporation difference in
the two SRM schemes as discussed above.
**3.2 Surface and tropospheric ozone response**

The ozone response is remarkably different in G4SSA and G4SSA-S.  Global mean

surface ozone increases under G4SSA-S and decreases under G4SSA, relative to RCP6.0 (Fig.
6a).  The total ozone column is dominated by stratospheric column ozone, and shows strong
reduction under G4SSA compared to RCP6.0, while the increase under G4SSA-S is small (Figs.
6b and 6d).  The underlying upward trends of total column ozone as well as stratospheric ozone
in all three scenarios are in line with the gradually declining stratospheric halogen content over
time (Figs. 6b and 6d).  As there is less halogen in the stratosphere towards the end of the
geoengineering, stratospheric ozone is recovering (Fig. 6d) and there is less reduction of the total
ozone column in G4SSA relative to RCP6.0 (Fig. 6b).  The agreement of all curves as concerns



interannual and decadal variations is because of the imposed QBO and 11-year solar cycles in all
the runs. The decreasing tropospheric ozone column and surface ozone after year 2060 in all
scenarios results from the global ozone precursor emissions decrease following the RCP6.0
scenario (Young et al., 2013).

The surface ozone concentration distributions in the three scenarios are similar, with the

highest concentration over the continents in the Northern Hemisphere (NH) (Fig. S5), while the
concentration differences between scenarios are spatially variable (Fig. 7). This highlights that
the key driver behind the absolute surface ozone abundances is the underlying ozone precursor
emissions following the RCP6.0 scenario. SRM is only a modulating factor, but depending on
the SRM scheme even the sign of its impact can differ; global mean surface ozone
concentrations in G4SSA are lower, relative to RCP6.0, whereas there are mild surface ozone
increases over the tropics (Fig. 7a). The strongest surface ozone reductions compared with
RCP6.0 occur in NH mid-latitudes across all seasons (Figs. S6a-d) and Southern Hemisphere
(SH) mid-to-high latitudes in MAM and JJA (Figs. S6b, c). As discussed in the next section, the
reduction over the continents is related to atmospheric chemistry changes while the reduction
over the polar regions in the winter hemisphere is due to transport. In G4SSA-S, surface ozone
also increases in the tropics relative to RCP6.0 (Fig. 7b), and this regional change is greater than
in G4SSA (Fig. 7c). Surface ozone decreases, however, are found at NH mid-latitudes over the
continents during all seasons (Fig. S6e-h). Comparing the two types of geoengineering
experiments directly, surface ozone concentrations are generally lower in G4SSA than in
G4SSA-S (Fig. 7c), with peak differences in terms of absolute changes (ppb) at SH mid-to-high
latitudes in MAM and JJA (Fig. S6i, j) and at NH mid-to-high latitudes in DJF (Fig. S6l).



### 3.3 Mechanisms of surface ozone change

Surface ozone concentrations are determined by chemical production and loss controlled by emissions of ozone precursors and the composition of the atmosphere, loss due to surface deposition of ozone, and the transport of ozone from other regions of the atmosphere (Monks et al., 2015). Since all scenarios considered here are based on the same RCP6.0 emission scenario of ozone precursors, such as nitrogen oxide ($NO_x$) and methane ($CH_4$), the differences in surface ozone must necessarily be driven by changes in climate in response to the geoengineering interventions, which include changes in temperature, humidity, atmospheric dynamics, and the photochemical environment. To understand the differences mechanistically, it is helpful to consider the impact of geoengineering on the tropospheric ozone budget.

The upper part of Table 1 shows the sources (production and net transport from the stratosphere (stratosphere-troposphere-exchange, STE)) and sinks (loss rates and dry deposition) of tropospheric ozone. Both G4SSA and G4SSA-S show positive net chemical change of tropospheric ozone (chemical production minus loss) and negative change in STE of ozone relative to RCP6.0. However, the magnitude of these changes is significantly different. Compared with RCP6.0, tropospheric ozone net chemical change increases by ~125 Tg yr$^{-1}$ and ~40 Tg yr$^{-1}$ in G4SSA and G4SSA-S, respectively, whereas STE of ozone decreases by ~140 Tg yr$^{-1}$ (~25%) and ~30 Tg yr$^{-1}$ (~5%) in G4SSA and G4SSA-S, respectively. The positive net chemical changes are the result of reductions in both chemical ozone production and loss under G4SSA and G4SSA-S relative to RCP6.0, with larger reductions in ozone loss reactions (Table 1). Specifically, G4SSA-S shows a ~90 Tg yr$^{-1}$ larger decrease in ozone chemical production, whereas ozone loss budgets are reduced by similar magnitudes for the two SRM schemes (262.5 Tg yr$^{-1}$ and 269.5 Tg yr$^{-1}$). Combining the chemical and transport changes, the tropospheric



ozone budget decreases under G4SSA and increases under G4SSA-S relative to RCP6.0, which
is consistent with the overall surface ozone changes.
The reasons for these specific changes are discussed in detail in the following two
sections.  Then, the impacts of the factors are combined to explain regional surface ozone
differences, as shown in Fig. 7.
**3.3.1 Chemical ozone production and loss in the troposphere**
Changes in tropospheric water vapor concentrations and the tropospheric photolysis
environment under G4SSA and G4SSA-S are key to understand the differences in tropospheric
ozone production and loss.  This result is consistent with results of a previous study for the case
of solar geoengineering under a more idealized forcing scenario (Nowack et al., 2016).
To explain this, we briefly re-iterate that tropospheric ozone ($O_3$) production is driven by
photolysis of nitrogen dioxide ($NO_2$) and the subsequent formation of ozone via a three-body-
reaction with resulting ground state atomic oxygen $O(^3P)$ (Monks, 2005),
$$NO_2 + h\nu \ (\lambda < 420 \ nm) \rightarrow NO + O(^3P) \tag{1}$$
$$O(^3P) + O_2 + M \rightarrow O_3 + M \tag{2}$$
where M is an inert collision partner (mostly molecular nitrogen).  $NO_2$ formation in turn is
crucially dependent on the oxidation of NO by reaction with peroxides present in the troposphere,
for example,
$$HO_2 + NO \rightarrow NO_2 + OH \tag{3}$$
$$RO_2 + NO \rightarrow NO_2 + RO \tag{4}$$
where R represents general organic residues such as $CH_3$ (row 6 in Table 1).  $RO_2$ in turn is
produced by oxidation reactions between VOCs and the hydroxyl radical OH.  Tropospheric OH





is formed primarily by ozone photolysis and subsequent reaction of excited atomic oxygen $O(^1D)$
with water vapor,

$$O_3 + h\nu \ (\lambda < 320 \ nm) \rightarrow O(^1D) + O_2 \qquad (5)$$

$$O(^1D) + H_2O \rightarrow 2 \ OH \qquad (6)$$

Reaction (6) competes with several other reactions due to the high reactivity of $O(^1D)$. However,
most importantly, the majority of $O(^1D)$ is quenched by collision with inert molecules such as
molecular nitrogen to ground state atomic oxygen $O(^3P)$, which subsequently undergoes
reformation to $O_3$ via three-body-reaction (2). Therefore, tropospheric ozone production and loss
is strongly linked to concentrations of water vapor and the photochemical environment
(availability of UV) in the troposphere.

In the case of clean (low $NO_x$) environments, lower water vapor concentrations (mainly

in the tropical region 30°N – 30°S) (Fig. S7) lead to less ozone loss via reactions (5) and (6) and
additional reactions with the formed HOx species (r-O(1D)-H2O, r-OH-O3, and r-HO2-O3 in
Table 1). This happens at the expense of more quenching of $O(^1D)$ and subsequent recycling of
ozone, thus resulting in ozone increases. In contrast, in polluted (high $NO_x$) environments, less
OH formation under lower atmospheric water vapor concentrations leads to reduced formation of
$RO_2$ and $HO_2$. Therefore, less $NO_2$ is produced via reactions (3) and (4), resulting in less
catalytic ozone production via reactions (1) and (2) (r-NO-HO2 and e.g. r-CH3O2-NO in Table 1).
Consequently, ozone production is reduced in $NO_x$-polluted environments under lower
atmospheric water vapor concentrations.

With these fundamentals in mind, it is possible to understand the sign of the tropospheric

ozone loss and production rate changes summarized in Table 1. Under both G4SSA and
G4SSA-S, the key objective is to achieve surface temperature decreases. Tropospheric water





vapor concentrations (or specific humidity) are strongly coupled to surface temperatures, because
relative humidity does not change much with climate change (Soden and Held, 2006; Dessler
and Sherwood, 2009), so that the surface cooling simultaneously reduces tropospheric specific
humidity by 5-20% depending on region and altitude. The slow-down of the hydrological cycle
under SRM will further enhance this tropospheric humidity reduction (Bala et al., 2002; Tilmes
et al., 2013; Nowack et al., 2016). As a result, less water vapor in both G4SSA and G4SSA-S
reduces ozone chemical loss by ~150 Tg $yr^{-1}$ through reactions (5) and (6). The resulting
decrease in $HO_x$ leads to further reductions in ozone loss, i.e., via reaction with OH (~20 Tg $yr^{-1}$)
and $HO_2$ (~60-70 Tg $yr^{-1}$). Overall, these water vapor/$HO_x$-related reactions explain ~90% of the
overall reduction in ozone loss under SRM compared to a future RCP6.0 simulation.

The reduction in atmospheric humidity also affects ozone production, but to a smaller

degree. Here, ozone production via reaction between NO and $HO_2$ is the key factor in driving
these changes, with reductions of ~55 and 120 Tg $yr^{-1}$ for G4SSA and G4SSA-S, respectively.
The signal of reduced OH production propagates through all other $NO_x$-catalyzed ozone
production pathways involving $RO_2$ via reactions (4) and subsequently (1) and (2). NO
oxidation via the $CH_3O_2$-NO pathway decreases by ~27 and 49 Tg $yr^{-1}$ in G4SSA and G4SSA-S.
Changes in natural $NO_x$ emissions by lightning play a minor role in comparison. In both SRM
schemes, the reduction of lightning induced $NO_x$ is not significant in most regions, and there is
no significant difference between the two SRM schemes (Fig. S8).

The changes in chemical ozone production rates tend to be smaller in the sulfate G4SSA

experiment than in the case of a solar constant reduction in G4SSA-S. There are three possible
factors that contribute to this:





1. The entire reaction cycle depends on the availability of sunlight to photolyse $O_3$ and
$NO_2$. Since SRM schemes modulate the intensity of sunlight (here by l%) reaching the
troposphere in order to mitigate tropospheric warming, this will necessarily also play a role in all
changes to ozone production and loss reactions in our SRM simulations. More importantly,
however, the sulfate injection geoengineering alters stratospheric ozone concentrations, which
ultimately impacts the photochemical environment of the troposphere by changing radiative
fluxes into the troposphere (DeMore et al., 1997; Nowack et al., 2016). For example, a reduced
stratospheric column will help to stimulate the tropospheric photochemistry by allowing more
radiation relevant reactions (1) and (5) to propagate into the troposphere.
2. Diffuse radiation under G4SSA promotes the photosynthesis rate and increases
canopy transpiration (Fig. S4). Therefore, we expect that water vapor concentration over the
continents with plants would be slightly higher in G4SSA relative to G4SSA-S (Fig. S9). Those
regions with higher water vapor (East Asia, South Asia, North America, South Africa) are
consistent with high $NO_x$ regions (Fig. S10). Therefore, the slightly smaller reduction of water
vapor under G4SSA in the regions above increases ozone chemical production compared with
G4SSA-S, and hence G4SSA shows less reduction of ozone chemical production than that in
G4SSA-S.
3. Different biogenic VOC emissions under G4SSA and G4SSA-S, which, due to their
central role in forming $NO_2$, are highly important for ozone production. In both scenarios, lower
temperatures reduce the heat stress on the emitting plants and therefore reduces their VOC
emissions (Tingey et al., 1980; Sharkey and Yeh, 2001; Lathière et al., 2005; Bornman et al.,
2015) (e.g., bio-emitted isoprene, Fig. S11). However, at the same time enhanced diffuse
radiation under G4SSA increases biogenic VOC emissions compared with G4SSA-S (Wilton et




al., 2011) (Fig. S11i, j, k and l). In Table 1, biogenic VOC-related ozone chemical production is
generally very similar between G4SSA with G4SSA-S (e.g., r-ISOPO$_2$-NO, r-MACRO$_2$-NOa, r-
MCO$_3$-NO and r-TERPO$_2$-NO), and contributes less than 2% to the overall difference between
G4SSA and G4SSA-S.
**3.3.2  Changes in stratosphere-troposphere exchange**
Stratospheric chemical and dynamical changes can impact tropospheric ozone not only
by changing the tropospheric photochemical environment, but also by changing the actual
transport of ozone from the stratosphere into the troposphere (Hegglin and Shepherd, 2009; Neu
et al., 2014). This can be either caused by changes in ozone concentrations in the stratosphere,
or by changes in the rate of exchange of air masses between stratosphere and troposphere (i.e.,
the strength of the Brewer–Dobson (B-D) circulation).
Fig. 8 shows seasonal latitude-height cross-sections of differences in ozone volume
mixing ratios between G4SSA and RCP6.0 as well as G4SSA-S and RCP6.0 for altitudes above
the 500 mb pressure level. Under G4SSA, heterogeneous reactions on the aerosol surfaces lead
to increased halogen activation and with that an enhancement of ozone depletion in mid to high
latitudes (Tilmes et al., 2008, 2009, 2012; Heckendorn et al., 2009). On the other hand,
heterogenous reactions reduce the NO$_x$ to NO$_y$ ratio, which results in an increase in ozone
mixing ratios, mainly in the middle stratosphere (Tie and Brasseur, 1995) (Fig. 8a). In addition,
changes in temperature also change the photochemistry of ozone. Altogether, this results in
year-round lower stratospheric ozone loss worldwide that peaks during the return of sunlight at
high SH latitudes. In comparison, the solar reduction in G4SSA-S does not enhance
stratospheric heterogeneous reactions. The much smaller change (increase) in ozone (Fig. 8b) is
driven by the change of homogeneous chemistry due to slight temperature reduction (Fig. 4b)



and the slowing of the B-D circulation under tropospheric cooling (Fig. 9b) (Lin and Fu, 2013;
Nowack et al., 2015; Shepherd and McLandress, 2011). The net result is small ozone increases
in the tropical lower stratosphere and decreases in both extratropical lower stratospheres (Fig.
8b).
Age of air is used to indicate the strength of the B-D circulation (Fig. 9). Here, it is
calculated relative to the zonal mean of 1°N at 158.1 mb (Garcia and Randel, 2008; Waugh,
2002). Older air indicates a slow-down of the B-D circulation. Compared with RCP6.0, both
G4SSA and G4SSA-S show older air in the stratosphere indicating a slowdown of the circulation.
The cooling effect in two SRM scenarios correlates with a weakening of tropical upwelling.
However, in G4SSA, the heating of the tropical stratosphere results in enhanced lifting, which
counteracts the weakening of the B-D circulation (Figs. 9a and 9c). Previous studies show
controversial results on how the B-D circulation changes due to extra aerosol in the atmosphere.
Aquila et al. (2012) modeled stronger tropic upwelling after the eruption of Mt. Pinatubo, and
other studies also found enhanced simulated B-D circulation after this volcano eruption (Aquila
et al., 2013; Pitari et al., 2016). However, with extra black carbon in the stratosphere, the
tropical upwelling weakens due to the simultaneous effect of tropospheric cooling (Shepherd and
McLandress, 2011; Mills et al., 2014).
The sum of both effects, stratospheric chemical changes and the impact of B-D
circulation change on STE of ozone, is shown in Fig. 10. In G4SSA, ozone transported from the
stratosphere to the troposphere is significantly decreased by ~25% relative to RCP6.0. In
G4SSA-S the reduction is small. Since the air mass transported from the stratosphere to the
troposphere is reduced in both scenarios, and is even more strongly reduced under G4SSA-S (Fig.
9), we find that the enhanced stratospheric ozone depletion in G4SSA is the dominant reason for



the strong reduction of STE of ozone. This is also supported by a stratospheric ozone tracer from
the model, $O_3^{Strat}$, which is set to ozone mixing ratios in the stratosphere and experiences only
chemical loss in the troposphere without chemical production (Fig. S12). . We thus conclude
that the significant changes in STE of ozone in G4SSA are mainly driven by enhanced
stratospheric ozone depletion catalysed through the aerosols (see also Table 1).

**3.3  Balance of the different mechanisms**

In summary, there are two main factors that determine the tropospheric ozone responses

in our SRM and RCP6.0 simulations: (a) changes in tropospheric ozone chemical production/loss
due to water vapor changes and impacts on the photochemical environment of the troposphere as
a result of both changes in stratospheric ozone and (to a smaller degree) the actual dimming of
sunlight depending on the geoengineering scheme, and (b) changes in stratosphere-troposphere
exchange of ozone.

These factors can also be used to explain the big picture behind the surface ozone

changes shown in Fig. 7. In G4SSA-S the reduced tropospheric humidity leads to stronger
reductions of ozone loss than the decreases in ozone production, leading to global increases in
surface ozone, but particularly in clean air oceanic environments in the tropics. This net increase
in ozone chemical change is not cancelled out by the slight reduction of ozone transport from the
stratosphere (Fig. 10). In G4SSA, reduction of ozone transport from the stratosphere is the
controlling factor, which overwhelms the increase in net ozone production. The effect is
particularly pronounced at mid-to-high latitudes (Fig. S12a), thus giving rise to surface ozone
decreases there (Fig. 7). In contrast, the effect of reduced tropospheric humidity is relatively
more important in the tropics than in other regions, which results in a local increase in surface
ozone under G4SSA. Regionally $HO_x$-$NO_x$ induced reductions in ozone production (Table 1)



can become important to explain surface ozone decreases in $NO_x$-polluted land areas in the NH
for both scenarios (Figs. 7 and S6). Further minor contributions to the differences in surface
ozone between G4SSA and G4SSA-S are a consequence of changes in water vapor due to
regional canopy transpiration effects and biogenic VOC emissions (e.g., isoprene, Table 1 and
Fig. S11).
**4. Conclusions**
Tropospheric ozone changes are to be expected in a geoengineered climate with
consequent impacts on air pollution and crop yields. However, for the scenarios considered here,
solar and sulfate geoengineering could have entirely different impacts, even in terms of the sign
of the response, a rare discrepancy for a surface signal between these two types of
geoengineering. There have been many studies using solar irradiance reductions to illustrate
SRM. However, it turns out that different SRM strategies would have different impacts on
hydrology, atmospheric dynamics, the terrestrial carbon sink, and as investigated in this paper,
tropospheric chemistry. These results also depend on the scenario of future ozone precursor and
halogen emissions.
We have identified and explained the mechanisms by which stratospheric sulfate
geoengineering would change surface ozone concentrations. We find that geoengineering might
have the potential to significantly reduce some climate impacts, but it cannot fix the problem of
air pollution. To reduce air pollution effectively, changes in surface emissions are key, with
changes in climate (including geoengineering) being only a modulating factor (Monks et al.,
2015; Stevenson et al., 2013; Young et al., 2013). More importantly, the surface ozone reduction
between 2030 and 2070 in G4SSA is primarily the result of decreased STE of ozone following
ozone depletion in the stratosphere. The rather mild pollution benefit under the RCP6.0



background would thus be bought at the expense of the delay of the stratospheric ozone
recovery, which would result in enhanced UV penetration to Earth's surface and with that
serious impacts on human health (e.g., skin cancer) and the ecosystem. In the future, potential
increases of stratospheric ozone as a result of geoengineering may result in an increase of surface
ozone, causing more ozone pollution. However, further analysis on air pollutants other than
ozone are needed.
As shown by Pitari et al. (2014), impacts on ozone from stratospheric geoengineering can
be highly model dependent. We consider the results here to be a GeoMIP testbed experiment,
and encourage others to compare our results to those from other climate models to evaluate the
robustness of the results presented here.

**Acknowledgments.** This work is supported by U.S. National Science Foundation (NSF) grants
AGS-1157525, GEO-1240507, AGS-1430051, and AGS-1617844. Computer simulations were
conducted on the National Center for Atmospheric Research (NCAR) Yellowstone
supercomputer. NCAR is funded by the NSF. The CESM project is supported by the NSF and
the Office of Science (BER) of the US Department of Energy. Peer Nowack is supported by the
European Research Council through the ACCI project, project number 267760. We thank Jean-
Francois Lamarque, Daniel Marsh, Andrew Conley, Louisa K. Emmons, Rolando R. Garcia,
Anne K. Smith, and Douglas E. Kinnison for the CAM4-Chem development.



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





**Table 1.** Tropospheric ozone production and loss rates (Tg yr$^{-1}$) over the period of years 2030-
2069 (average of three ensemble members).  For chemical ozone production and ozone loss the
net impacts of only the most important reaction pathways are listed.

|  | RCP6.0 | G4SSA | G4SSA-S |
|---|---|---|---|
| $O_3$ Net Chemical Change | 346.1 | 472.7 | 384.8 |
| $O_3$ Tropospheric Dry Deposition | 901.5 | 891.5 | 909.4 |
| $O_3$ STE* | 555.4 | 418.8 | 524.6 |
| $O_3$ Production | 4895.8 | 4764.0 | 4671.8 |
| r-NO-HO$_2$ | 3087.3 | 3031.0 | 2964.8 |
| r-CH$_3$O$_2$-NO | 1132.3 | 1105.2 | 1083.1 |
| r-PO$_2$-NO | 21.8 | 20.1 | 19.9 |
| r-CH$_3$CO$_3$-NO | 183.1 | 172.2 | 171.2 |
| r-C$_2$H$_5$O2-NO | 6.6 | 6.7 | 6.7 |
| 0.92*r-ISOPO$_2$-NO | 149.8 | 135.3 | 134.0 |
| r-MACRO$_2$-NOa | 76.1 | 69.8 | 69.5 |
| r-MCO$_3$-NO | 34.5 | 30.5 | 30.3 |
| r-RO$_2$-NO | 12.2 | 11.5 | 11.5 |
| r-XO$_2$-NO | 66.5 | 60.8 | 60.5 |
| 0.9*r-TOLO$_2$-NO | 4.1 | 4.1 | 4.1 |
| r-TERPO$_2$-NO | 18.1 | 16.9 | 16.8 |
| 0.9*r-ALKO$_2$-NO | 22.9 | 23.0 | 22.9 |
| r-ENEO$_2$-NO | 12.5 | 11.6 | 11.7 |
| r-EO$_2$-NO | 36.8 | 34.6 | 34.5 |
| r-MEKO$_2$-NO | 17.7 | 17.9 | 17.8 |
| 0.4*r-ONITR-OH | 7.5 | 6.8 | 6.8 |
| r-jonitr | 1.4 | 1.2 | 1.2 |
| $O_3$ Loss | 4421.1 | 4158.6 | 4151.6 |
| r-O(1D)-H$_2$O | 2430.4 | 2286.5 | 2263.5 |
| r-OH-O$_3$ | 548.2 | 528.3 | 527.0 |
| r-HO$_2$-O$_3$ | 1288.9 | 1216.7 | 1232.9 |
| r-C$_3$H$_6$-O$_3$ | 13.8 | 11.5 | 11.5 |
| 0.9*r-ISOP-O$_3$ | 71.4 | 58.0 | 57.6 |
| r-C$_2$H$_4$-O$_3$ | 9.3 | 7.8 | 8.0 |
| 0.8*r-MVK-O$_3$ | 18.6 | 15.5 | 15.7 |
| 0.8*r-MACR-O$_3$ | 3.5 | 2.9 | 2.9 |
| r-C$_{10}$H$_{16}$-O$_3$ | 37.0 | 31.5 | 31.6 |

*$O_3$ STE is ozone transported through the Stratosphere Troposphere Exchange.  We calculated
this value using equation –
$$O_{3\ STE} + O_{3\ net\ tropospheric\ chemical\ change} + O_{3\ dry\ tropospheric\ deposition} = 0$$
Tropospheric ozone is defined as ozone concentration less than 150 ppb.





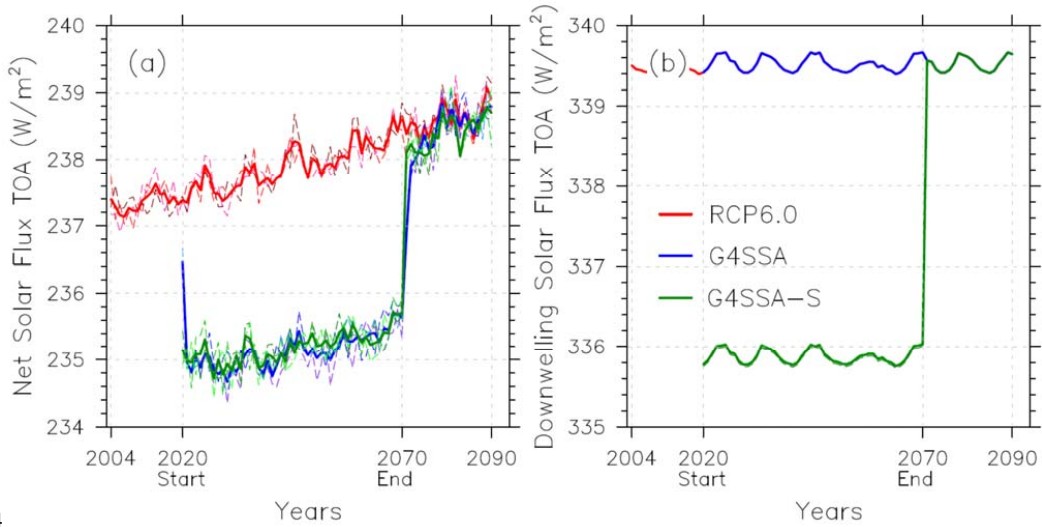


**Figure 1.** (a) Global averaged annual net solar flux on the top of the atmosphere (W/m²) and (b)
downwelling solar flux on the top of the atmosphere (W/m²). Dashed lines are ensemble
members, and solid lines are the average of three ensemble members. Geoengineering starts at
January 1st 2020 and ends at January 1st 2070. The 11-year periodicity is imposed as a prediction
of the sunspot cycle. In (b) the G4SSA curve exactly covers the RCP6.0 curve.



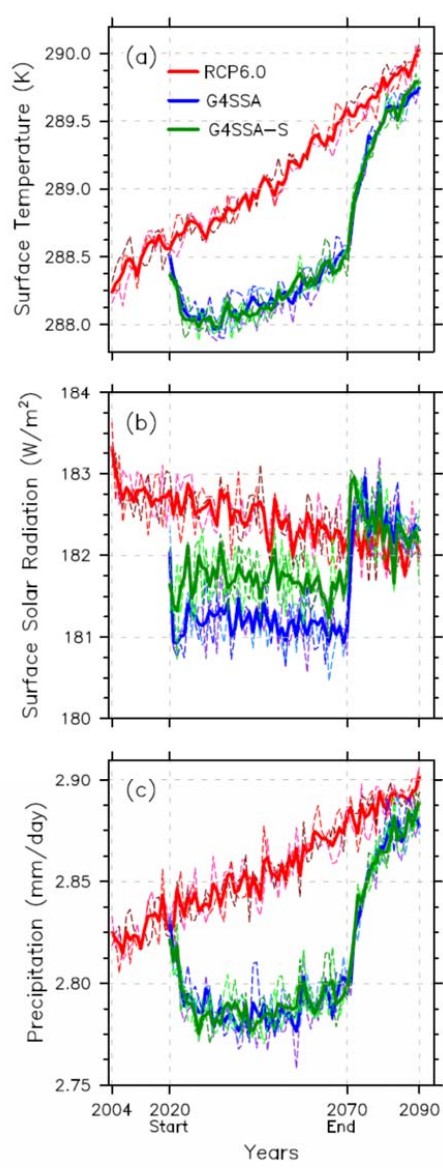

691
**Figure 2.** (a) Global averaged annual surface air temperature (K), (b) downwelling surface solar
radiation (W/m$^2$), and (c) Land average annual canopy transpiration (mm/day). Dashed lines are
ensemble members, and solid lines are the average of the three ensemble members.
Geoengineering starts at 1 January 2020 and ends at 1 January 2070.





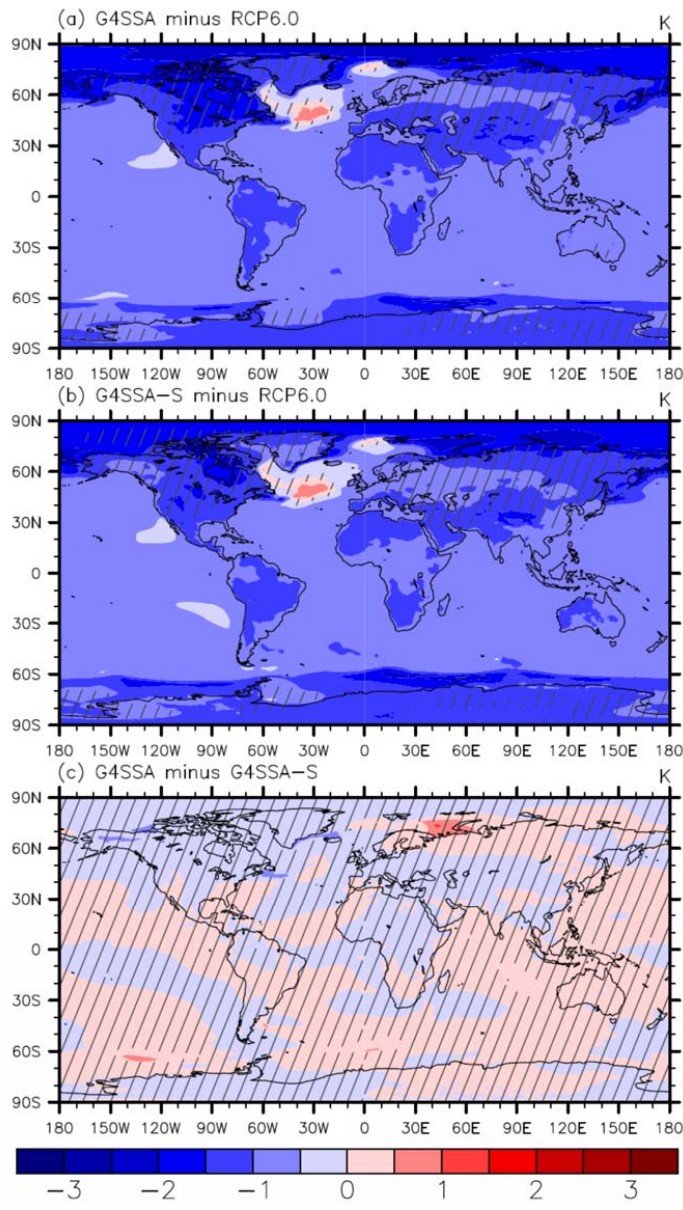

**Figure 3.** Global maps of surface temperature differences (K) between (a) G4SSA and RCP6.0,
(b) G4SSA-S and RCP6.0, and (c) G4SSA and G4SSA-S over the period 2030-2069. Hatched
regions are areas with $p > 0.05$ (where changes are not statistically significant based on a paired
$t$-test).



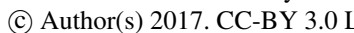




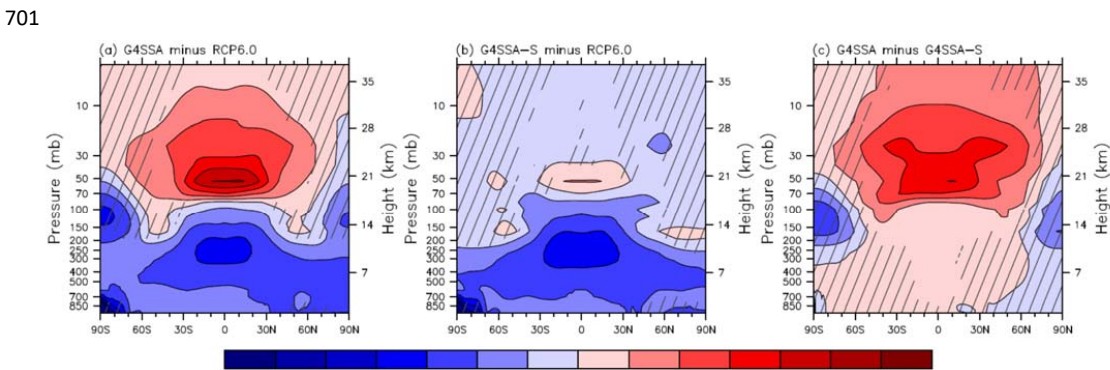

70_

**Figure 4.** Zonal mean temperature differences (K) in the geoengineering experiments (a)
G4SSA minus RCP6.0, (b) G4SSA-S minus RCP6.0, and (c) G4SSA minus G4SSA-S. These
are averaged for three ensemble members for years 2030-2069. Hatched regions are
insignificant, with $p > 0.05$.





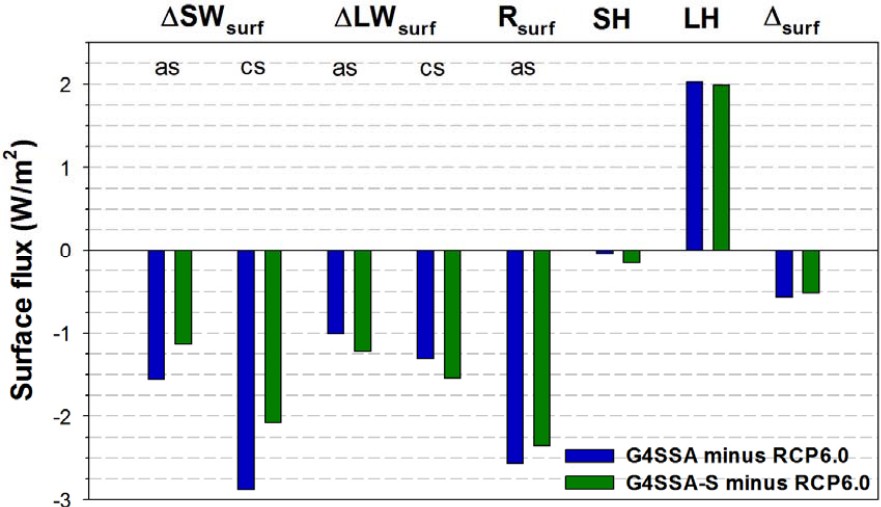

**Figure 5**. Energy flux at the surface, shown as G4SSA minus RCP6.0 and G4SSA-S minus
RCP6.0 for 2030-2069. For all fluxes, downwelling is positive. $\Delta SW_{surf}$ is the net shortwave
flux at the surface, $\Delta LW_{surf}$ is the net longwave flux at the surface, $R_{surf}$ is the sum of $\Delta SW_{surf}$ and
$\Delta LW_{surf}$, SH is sensible heat, LH is latent heat, and $\Delta L$ is the sum of $\Delta SW_{surf}$, $\Delta LW_{surf}$, SH, and
LH; as is all sky and cs is clear sky.



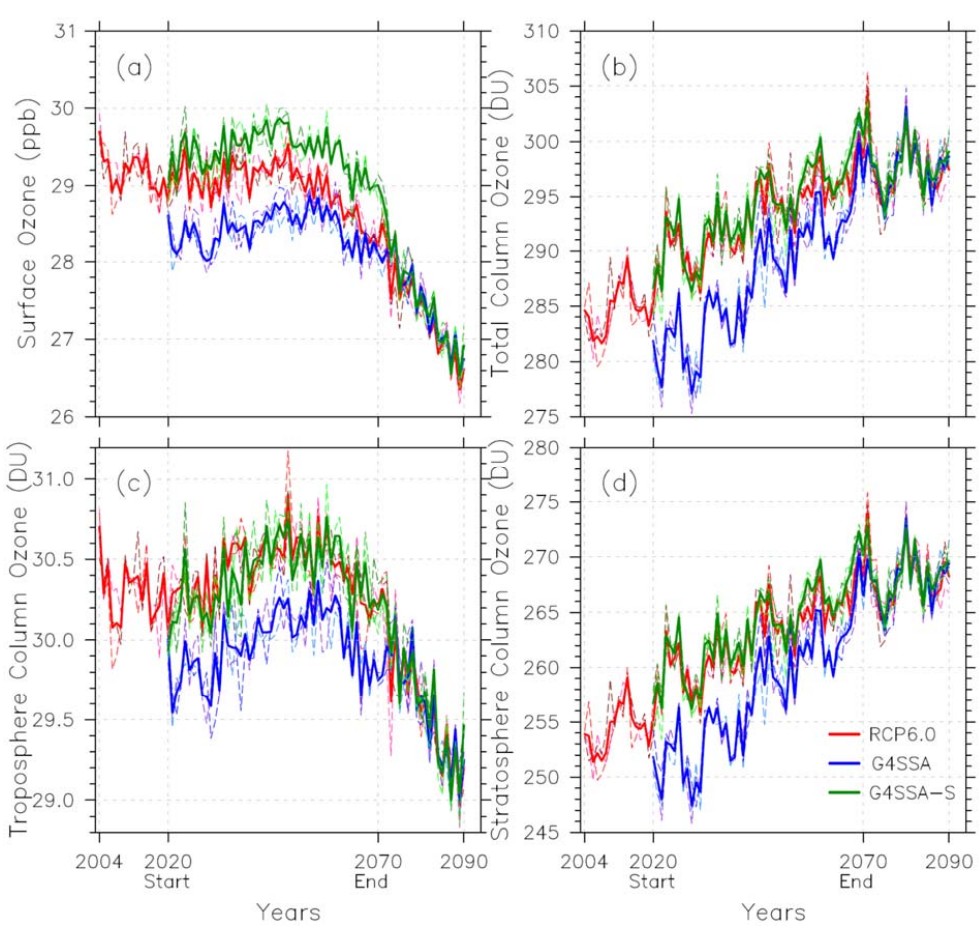

**Figure 6.** (a) Global averaged annual surface ozone concentrations (ppb), (b) total column
ozone (DU), (c) tropospheric column ozone (DU), and (d) stratospheric column ozone (DU).
Ozone concentration of 150 ppb is used as the boundary of tropospheric ozone and stratospheric
ozone. Dashed lines are ensemble members, and solid lines are the average of the three
ensemble members. Geoengineering starts at 1 January 2020 and ends at 1 January 2070.





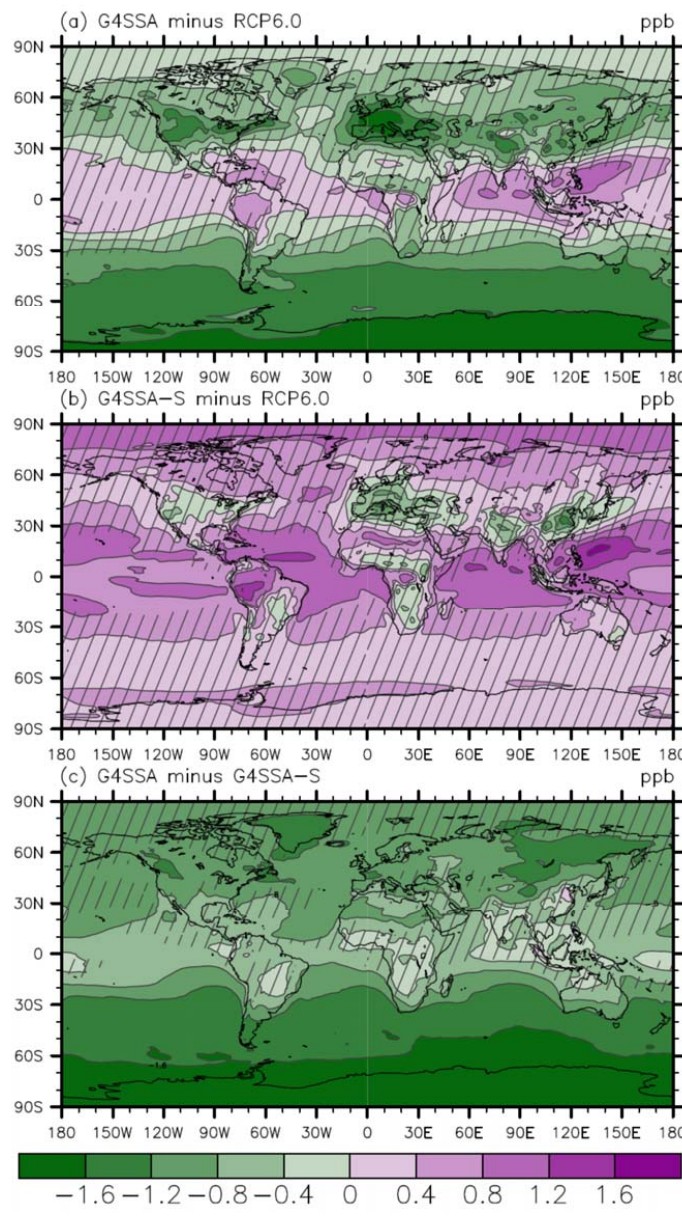


**Figure 7**. Global maps of surface ozone concentration differences (ppb) between (a) G4SSA
and RCP6.0, (b) G4SSA-S and RCP6.0, and (c) G4SSA and G4SSA-S for 2030-2069. Hatched
regions are insignificant, with $p > 0.05$.






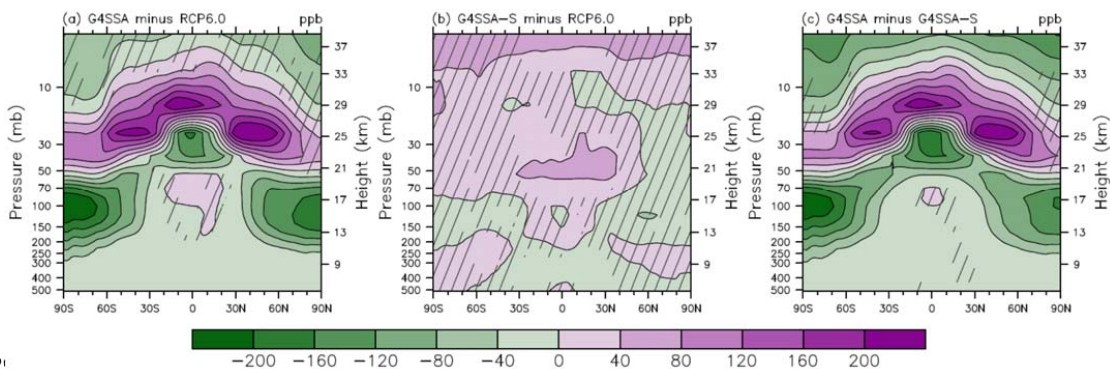

725

**Figure 8.** Zonal mean ozone concentration differences (ppb) in the geoengineering experiments, averaged for three ensemble members for 2030-2069. Hatched regions are insignificant, with $p >$ 0.05.




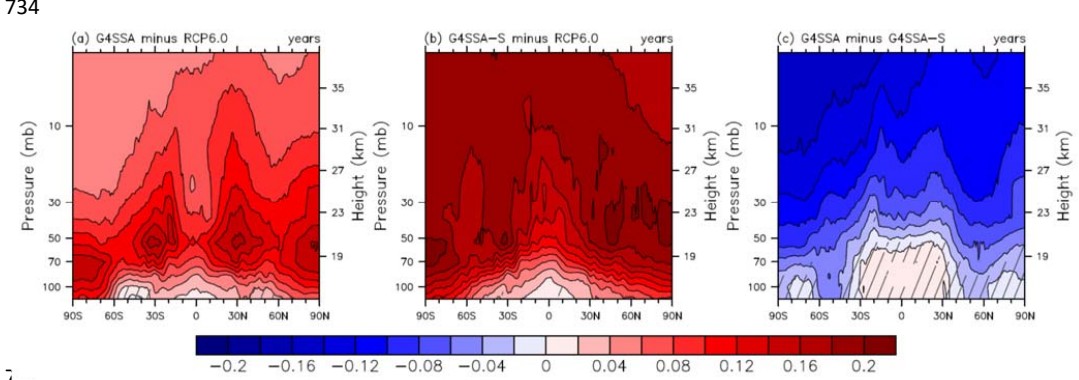

**Figure 9.** Zonal mean age of air differences (years) between (a) G4SSA and RCP6.0, (b)
G4SSA-S and RCP6.0, and (c) G4SSA and G4SSA-S. They are averaged for three ensemble
members for 2030-2069. Hatched regions are insignificant, with $p > 0.05$.





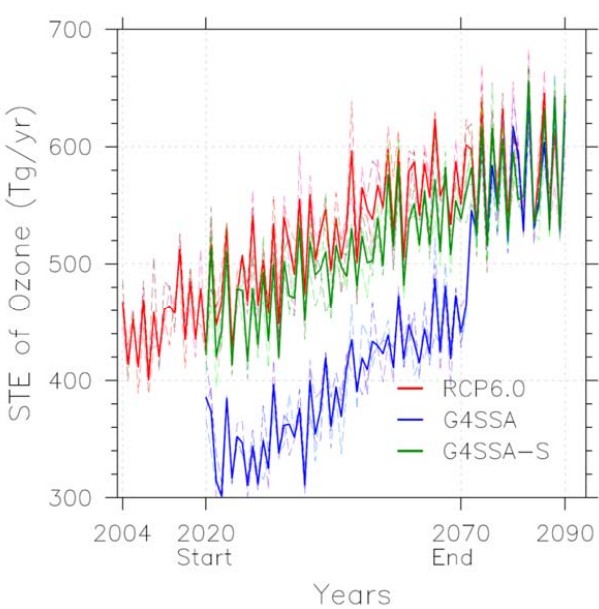


**Figure 10.** Global annual averaged ozone transported from the stratosphere to the troposphere
(STE of ozone) in Tg yr$^{-1}$. Geoengineering starts at 1 January 2020 and ends at 1 January 2070.