# Peer review of "Impacts of Stratospheric Sulfate Geoengineering on Tropospheric Ozone"

_Atmospheric Chemistry and Physics, 2017_

## Referee Comment (RC1) · U. Niemeier (Referee) · 6 Jun 2017

Review on:
**Impacts of Stratospheric Sulfate Geoengineering on Tropospheric Ozone
(acp-2017-434)**
Lili Xia et al

June 6, 2017
Xia et al describe the impact of two solar radiation management (SRM) techniques, stratospheric sulfur injections and solar dimming, on tropospheric and stratospheric ozone concentrations. They use a Earth-System model coupled to a chemistry and a

land model. With this model setup they are able to simulate not only the impact of SRM on ozone but also the relation to plants.

Only few geoengineering studies put the focus on ozone changes. Changes in ozone are an important side effect of SRM. Therefore, this paper helps in further understanding possible consequences and side effects of SRM. The paper is well written and I support publication after addressing the following minor remarks.

**General:**

Different to the GeoMIP experiments G1 and G3, where greenhouse gas forcing is counteracted by balancing the top of the atmosphere (TOA) imbalance, the two techniques in this paper are balanced by the solar TOA forcing. However, they compare the results to Ferraro et al (2014) and Niemeier et al (2013) but these studies used the TOA imbalance. The paper is missing a discussion of this aspect. It might be an option to apply a bias correction, following Niemeier et al (2013). At least the discussion on precipitation should include the TOA energy balance and follow Liepert and Prevedi (2009) (see also Eq 8 in Niemeier et al 2013).

The naming of the experiments is sometimes confusing for the reader. Both SRM techniques start with S. G4SSA and G4SOL could be an alternative.

**Specific comments:**

Line 119: The number of vertical levels is quits small compared to the horizontal resolution.

Line 128: Can you discuss the possible impact of the reduces solar radiation on the photolysis rate.

Line 169: This sentence is miss leading. It sounds like all references warm by 3 K.

Line 170: G4SSA-S shows also a slight warming in the lower stratosphere. Why?

Line 178 to 183: TOA imbalance as mentioned above. The difference between R-toa and R-surf are compensate by condensational heating.
Eq 1 in Niemeier et al (2013) was used for a bias correction. It might be an option to use this correction (G4SSA-S for I-FIX) to compensate the solar balancing but G4 has still a transient climate. However, it might be worth trying.

Line 187: 'They attribute...' better 'Ferraro attribute ..' This was described earlier in a paper by Bala et al (2008).

Line 222: Adding a row with differences in percent to the plot (Fig 7).

Line 324: You say earlier that you do not change the photolysis rate. How is the the reduced sunlight changing ozone here?

Line 356: This is not the only way to exchange ozone between stratosphere and troposphere. STE due to tropopause folds in the surf zone might be more important.

Line 383: Have you mentioned black carbon before?

Line 373 to 385: Your study uses a fixed QBO, different to Aquilla et al (2012). This may play an important role. Niemeier and Schmidt (2017) show also an increase in vertical velocity as well as a strong impact on stratospheric transport. This aspect cannot be taken into account in this study but might be discussed as a caveat.

**Figures**

May explanations in the text base on figures in the appendix. You may add them, or some of them, in the main text. It is easier for the reader and does not really matter in an online paper.

Fig 2: Plot the ensemble mean as a running mean. This helps in recognizing differences or similarities.

Fig 4 and fig 8: It would be nice to see the position of the sulfate aerosol as a contour line.

Fig 5: Add TOA energy fluxes

Fig 6: Add additional years to the x-axis. Ozone decreases in RCP6 from roughly 2050, in G3SSA-S as well but not in G4SSA. Do you have an explanation?

Fig 7: Add a row with differences in % (or use the figure from the appendix).

**References**

Bala et al (2008): doi:10.1073/pnas.0711648105

Liepert and Prevedi (2009), J.Climate, http://dx.doi.org/10.1175/2008JCLI2472.1

Niemeier and Schmidt (2017): ACPD, https://doi.org/10.5194/acp-2017-470

---

## Referee Comment (RC2) · Anonymous Referee #2 · 7 Jul 2017

On the whole, this paper is well-written except for a few places where clarification is needed. For one, the description of the formulation of stratospheric chemistry in CAM-chem is missing. I was also wondering if the halogen loading and GHG concentrations all follow RCP 6.0 specifications such that the only differences between these three ensemble runs are sulfate aerosol loading for G4SSA and reduced solar fluxes for G4SSA-S. Regardless, the time series of halogen loading should be given at some point in section 2. Citation of other research papers alone won't do for a broader readership.

Other minor points, (1) Lines 75-76: Please explain how emission pathways can determine transport from the ozone-rich stratosphere. (2) Lines 304-305: Can you be more specific on how " the slow-down hydrological cycle under SRM will further enhance this

tropospheric humidity reduction"?

Lastly for the geoengineering assessment to be practical, there needs to be a specific metric for measuring the impact. For example, the change in the distribution of tropospheric ozone in terms of the probability density distribution of surface ozone concentration might be useful. And the authors need to discuss what can be improved in the modeling effort in the discussion section.

---

## Referee Comment (RC3) · Anonymous Referee #3 · 23 Jul 2017

This manuscript examined the effects of stratospheric sulfate aerosol and solar insolation reduction on tropospheric ozone and surface ozone. The study also examines the both chemical and transport mechanisms of tropospheric ozone changes to SRM techniques. The findings of this paper help us get a better understanding of effects of SRM. In general, I found the main points and the structure of this manuscripts are clear. Below are my comments for making the manuscript more concise. I recommended the paper to be published with minor revision.

Detailed comments:

Line 54: Add one or two references of sulfate aerosol effects in the stratosphere.

Caption for Figure S1: What do you mean about the 10N to 10S gridded present day

[Figure]

MLS/OMI satellite data? Doesn't the data cover the extra-tropics?

Line 159: move "the last 40 years of geoengineering" in line 161 to here.

Line 163: Figure 3a and 3b: why there is significant temperature increase over north Atlantic? More explanations of temperature changes between SRM runs and RCP 6.0 would be helpful.

Line 164-166: In Figure 3c: I did not see clear warming signal in Asia. And 'warming' here is confusing: both G4SSA and G4SSA-S show temperature decrease compared to RCP6.0. The red color in Figure 3c just means that G4SS4 has less temperature reduction than G4SS4-S.

In Figure S3, surface temperature in G4SS4 does show a significant warming over northern Europe and Asia compared to RCP6.0 in winter. I think that is the feature that agrees with the characteristic "winter warming" from volcanic stratospheric aerosol (Robock, 2000).

Line 168-170: around 50 hPa in the tropics, the G4SSA-S also shows the significant warming.

Line 173: would be better to switch the sequence of Figure 2b and Figure 2c

Line 207: How the halogen changes over these three runs?

Line 361: mid-high latitude: are you talking about the lower stratosphere $\sim$ 100 hPa? Adding the location of tropopause in Figure 8 would be much helpful.

Line 364: temperature changes in which direction?

Line 364: "Altogether, this results in year-round lower stratospheric ozone loss worldwide that peaks during the return of sunlight at high SH latitudes." Which figure describes this feature?

Line 368: Figure 8b: Confused here: in Figure 4b, temperature shows an increase

in the tropics around 50-70 hPa. Other regions show temperature reduction. You mentioned that ozone increase in Figure 8b is due to temperature decreases. While the regions with T increases (tropics, 50-70 hPa), ozone has a maximum increase. It would be much easier to understand if you mention lat/pressure when describing figures. Lower stratosphere in the polar region could reach as low as 400 hPa. Adding a tropopause in Figure 8 would be much helpful.

Line 383-384: Why tropical upwelling response differently between this study and Aquila et al 2012? Please be more specific?

Line 394: (Fig. S12). . delete one period

---

## Author Comment (AC1) · 22 Aug 2017

**Reply to comments on "Impacts of Stratospheric Sulfate Geoengineering on Tropospheric Ozone" by Xia, Nowack, Tilmes and Robock, submitted to *Atmospheric Chemistry and Physics*.**

*Comments are repeated in black italics.* Replies are indicated in blue.

**Reviewer #1**

*General comments:*

*Different to the GeoMIP experiments G1 and G3, where greenhouse gas forcing is counteracted by balancing the top of the atmosphere (TOA) imbalance, the two techniques in this paper are balanced by the solar TOA forcing. However, they compare the results to Ferraro et al (2014) and Niemeier et al (2013) but these studies used the TOA imbalance. The paper is missing a discussion of this aspect. It might be an option to apply a bias correction, following Niemeier et al (2013). At least the discussion on precipitation should include the TOA energy balance and follow Liepert and Prevedi (2009) (see also Eq 8 in Niemeier et al 2013).*

Thanks for pointing that out. Niemeier et al. (2013) performed a FIX scenario, which freezes greenhouse gas concentrations. FIX is treated as an analog for a perfect compensation of the forcing through the greenhouse gas increase after 2020 and is unbiased by construction. They then bias corrected the mean net flux imbalance at TOA for three different SRM scenarios (sulfate injection, marine cloud brightening and solar reduction) to be the same as in FIX. After that correction, the global averaged surface temperature follows very similar trajectories in all scenarios. In our simulation, we keep the net solar flux at the TOA the same in G4SSA and G4SSA-S.

Although our experimental designs are different compared to Niemeier et al. (2013), we find that the resulting effects on the TOA fluxes are quite similar in G4SSA and G4SSA-S: the two geoengineering scenarios show similar TOA fluxes reduction (we have added Fig. S5 and Fig. 5a). Therefore, we think that bias correction is not needed, and differences in the hydrological cycle response are rather caused by differences in the model set-ups (ozone chemistry, lapse rate, evapotranspiration) as is already mentioned in the manuscript.

In order to highlight these differences in the experimental approach, the text in this section of the paper now reads (lines 197-230):

"The similar evaporation reduction in G4SSA and G4SSA-S can also be explained by the surface energy budget (Fig. 5b). Although we keep the net shortwave radiation at the TOA the same in the two schemes (Fig. 1a and Fig. 5a), surface net solar radiation reduces more in G4SSA than in G4SSA-S (Fig. 2c and Fig. 5b) due to the absorption by sulfate aerosols in the near-infrared. This stronger surface solar forcing in G4SSA-S is mainly balanced by larger net longwave radiation to the atmosphere (Fig. 5). As a result, latent heat changes in the two scenarios are similar.

Here, precipitation and evaporation changes are very similar under sulfate and solar geoengineering. This is different from previous studies by Niemeier et al. (2013) and Ferraro et al. (2014) who found that the effect on the hydrological cycle is larger for

sulfate geoengineering. These differences are related to the experimental design. Niemeier et al. (2013) bias corrected all geoengineering scenarios to keep the net total flux at the TOA the same as that in 2020, while we keep the same net solar flux at the TOA in G4SSA and G4SSA-S (Fig. 1a). However, we found the net total fluxes at the top of the model in G4SSA and G4SSA-S are similar as well (Fig. 5a and Fig. S5). Therefore, differences in the TOA boundary conditions might not be the main reason for the different hydrological cycle responses. In their studies, with the same magnitude of surface cooling, the sulfate injection scenario led to a greater reduction of globally averaged evaporation and precipitation as compared with the solar case. Ferraro et al. (2014) attributed the enhanced hydrological cycle response to sulfate geoengineering to extra downwelling longwave radiation because of stratospheric heating from the injected aerosols. Sulfate geoengineering thus led to a relative stabilization of the troposphere (by heating the upper troposphere more than the mid-lower troposphere) compared with the solar reduction case (which we do not find, Fig. 4c). A more stratified troposphere, in turn, results in a stronger reduction of latent heat fluxes and precipitation (similar to theoretical considerations by Bala et al. (2008)). We find two possible reasons for the different response in our experiments. (1) The column ozone change could play an important role. In Niemeier et al. (2013) and Ferraro et al. (2014), the same prescribed ozone was used in all scenarios, while we used a fully coupled atmosphere-chemistry model. As shown in section 3.2, total column ozone in G4SSA reduces by about 5 DU (mainly in the lower stratosphere) compared with RCP6.0 and G4SSA-S (Fig. 6). Less ozone in G4SSA will change its radiative forcing, surface radiative fluxes and atmospheric lapse rate (Chiodo and Polvani, 2015; MacIntosh et al., 2016; Nowack et al., 2015, 2017) and thus contribute to the differences between the two studies. (2) Enhanced transpiration in G4SSA due to enhanced diffuse radiation reduces the evaporation difference in the two SRM schemes as discussed above."

*The naming of the experiments is sometimes confusing for the reader. Both SRM techniques start with S. G4SSA and G4SOL could be an alternative.*

Thanks, but we think G4SSA-S indicates solar reduction with the same forcing of G4SSA. G4SOL would be confused with solar reduction with the same forcing as G4.

*Specific Comments:*
*Line 119: The number of vertical levels is quits small compared to the horizontal resolution.*

Yes, this is CAM4-chem – a low top atmosphere model with fairly low vertical resolution. We understand the constraint of low vertical resolution, which might have an effect on the simulation of stratospheric dynamics. However, the representation of ozone in CAM4-chem has been evaluated and compares well to observations (Tilmes et al., 2016), as also shown in Fig. S1 and Fig. S2 in the supplement. In addition, the STE of ozone found here lies well within the range of typical model values in (high-top and high resolution) chemistry-climate models (Young et al., 2013). In future work, we plan to analyze output from a sulfate geoengineering simulation from the high top model WACCM to further investigate stratospheric dynamical changes with injected sulfate aerosols.

Young et al.: Pre-industrial to end 21st century projections of tropospheric ozone from the Atmospheric Chemistry and Climate Model Intercomparison Project (ACCMIP), Atmos. Chem. Phys., 13, 2063-2090, doi:10.5194/acp-13-2063-2013, 2013.

*Line 128: Can you discuss the possible impact of the reduces solar radiation on the photolysis rate.*

Since the reviewer pointed the question is related to "Interaction between aerosol burden and photolysis rates is not included in the model", we assume that the reviewer wants more details between the solar reduction due to aerosol and photolysis rate. We have added to the discussion in lines 137-142:

"Changes in photolysis rates in the troposphere depend on the stratospheric ozone column change (Kinnison et al., 2007). Increased ozone depletion as the result of geoengineering would therefore leads to an increase in UV in mid- and high latitudes. Since our model does not include the aerosol scattering effect on UV, expected UV reductions from the increased sulfate aerosol layer is not taken into account, which might result in an overestimation the tropospheric photolysis."

*Line 169: This sentence is miss leading. It sounds like all references warm by 3 K.*

We have changed it in Lines 185-187:

"Sulfate aerosol in the stratosphere results in strong warming by 3 K in the tropics (Fig. 4a), while in G4SSA-S there is slight cooling (Fig. 4b), consistent with previous studies (Tilmes et sl., 2009; Ammann et al., 2010; Jones et al., 2011). "

*Line 170: G4SSA-S shows also a slight warming in the lower stratosphere. Why?*

The slight warming in the lower stratosphere is probably a result of slightly increasing of ozone concentration in that regions and dynamical heating as the Brewer Dobson circulation slowing down (Fig. 8b). We have added in Lines 187-189:

"The slight warming in the lower stratosphere under G4SSA-S (Fig. 4b) might be a result of ozone changes and dynamical heating (discussion in Section 3.3.2)."

*Line 178 to 183: TOA imbalance as mentioned above. The difference between R-toa and R-surf are compensate by condensational heating. Eq 1 in Niemeier et al (2013) was used for a bias correction. It might be an option to use this correction (G4SSA-S for I-FIX) to compensate the solar balancing but G4 has still a transient climate. However, it might be worth trying.*

In our simulation, we keep the net solar flux at the TOA the same in G4SSA and G4SSA-S, which results in the same net total fluxes change at the top of the model (Fig. S5 and Fig. 5a). Therefore, it is not necessary to bias correct the two scenarios as surface temperatures are also essentially identical (within expected limits). We have added text in the manuscript to highlight this. Please see our response to the general comment on page 1.

*Line 187: 'They attribute...' better 'Ferraro attribute ..' This was described earlier in a paper by Bala et al (2008).*

Bala et al. (2008) has been added. Changed.

*Line 222: Adding a row with differences in percent to the plot (Fig 7).*

We have added the differences in percentage as Fig. S7.

*Line 324: You say earlier that you do not change the photolysis rate. How is the reduced sunlight changing ozone here?*

In the section "Model and Experiment Design," we mentioned that interactions between aerosol burden and photolysis rates are not included in the model and we have added more details (Lines 128-134). Changes in ozone will be triggered due to the changes in sunlight via (a) reduced ozone photolysis and (b) reduced oxygen photolysis that is needed for ozone production. Changes in stratospheric ozone in turn will affect tropospheric photolysis rates. However, overall such solar dimming effects will be small relative to other effects changing ozone (e.g., temperature and humidity) as the dimming in G4SSA-S is equivalent only to about 1% solar constant reduction.

*Line 356: This is not the only way to exchange ozone between stratosphere and troposphere. STE due to tropopause folds in the surf zone might be more important.*

Added in Lines 384-385:

"the rate of exchange of air masses between stratosphere and troposphere (i.e., the strength of the Brewer–Dobson (B-D) circulation and tropopause folds)."

*Line 383: Have you mentioned black carbon before?*

No, we didn't mention black carbon before. The only reason we put it here is stratospheric black carbon reduces tropospheric temperature and warms up stratosphere, similar to sulfate. And in those two studies, they showed a slow-down of tropical upwelling.

*Line 373 to 385: Your study uses a fixed QBO, different to Aquilla et al (2012). This may play an important role. Niemeier and Schmidt (2017) show also an increase in vertical velocity as well as a strong impact on stratospheric transport. This aspect cannot be taken into account in this study but might be discussed as a caveat.*

Thanks for pointing out that. Since Niemeier and Schimidt (2017) is currently ACPD, we will add this reference if it publishes before the publication of this manuscript. To strengthen the argument, we have added:

Lines 416-419:

"The differences between previous studies and our result may be because some previous studies used fixed ozone, with different stratospheric heating rates. In addition, in previous studies, the QBO was interactively simulated and the models had a higher model top."

*Figures*

*May explanations in the text base on figures in the appendix. You may add them, or some of them, in the main text. It is easier for the reader and does not really matter in an online paper.*

The paper already has 10 figures, and there are 15 more in the supplemental. We feel that we present the research well already with the 10 figures, and adding a couple more would not make much difference. 10 figures is the traditional number of figures for a paper. The supplemental figures are for those interested in more details.

*Fig 2: Plot the ensemble mean as a running mean. This helps in recognizing differences or similarities.*

The ensemble means are already quite smooth curves, and a running mean would make it harder to see the changes at times of implementation and termination of geoengineering.

*Fig 4 and fig 8: It would be nice to see the position of the sulfate aerosol as a contour line.*

Done

*Fig 5: Add TOA energy fluxes*

Done

*Fig 6: Add additional years to the x-axis. Ozone decreases in RCP6 from roughly 2050, in G3SSA-S as well but not in G4SSA. Do you have an explanation?*

Axis changed. We are not sure which panel of Fig. 6 you refer to. In all the panels, the ozone depletion in the stratosphere in G4SSA is evident compared to the other two scenarios. The differences in surface ozone are explained in the paper.

*Fig 7: Add a row with differences in % (or use the figure from the appendix).*

Done. Figure added as Figure S7.

**Reviewer #2**

*General comments:*
*On the whole, this paper is well-written except for a few places where clarification is needed.*

Thank you!

*For one, the description of the formulation of stratospheric chemistry in CAM- chem is missing.*

We have added in Lines 128-134:

"The tropospheric chemical mechanism in CAM4-chem is based on the Model for Ozone and Related chemical Tracers (MOZART), version 4 (Emmons et al., 2010). The stratospheric chemical mechanism is described in Kinnison et al. (2007), Lamarque et al. (2012) and Tilmes et al. (2015), and the complete chemical reactions included (photolysis, gas-phase chemistry and heterogeneous chemistry) are listed in Tilmes et al. (2016b), Table A2. Reaction rates are updated following Jet Propulsion Laboratory 2010 recommendations (Sander et al., 2011)."

*I was also wondering if the halogen loading and GHG concentrations all follow RCP 6.0 specifications such that the only differences between these three ensemble runs are sulfate aerosol loading for G4SSA and reduced solar fluxes for G4SSA-S.*

Yes, that is correct. In all scenarios (G4SSA, G4SSA-S and RCP6.0), the anthropogenic emissions follow the same pathway, and the only difference is the prescribed sulfate injection in G4SSA and the solar constant reduction in G4SSA-S. We have added a sentence to make this clear.

Lines 151-152: "Both geoengineering scenarios include RCP6.0 forcings."

*Regardless, the time series of halogen loading should be given at some point in section 2. Citation of other research papers alone won't do for a broader readership.*

We have added in Lines 238-239:

"The halogen loading in the three scenarios is the same, and more information can be found in Morgenstern et al. (2017)."

The halogen loading is in Figure 1(b) of Morgenstern et al. (2017).

*Other minor points, (1) Lines 75-76: Please explain how emission pathways can determine transport from the ozone-rich stratosphere.*

Different emission pathways of halogens determine the recovery period of the ozone hole. The stratospheric ozone concentration will partially determine the ozone transported from stratosphere to troposphere. In addition, greenhouse gas emission pathways will result in different levels of global warming, which changes the stratospheric dynamics, such as the Brewer-Dobson circulation, and therefore alter the ozone transported from stratosphere to troposphere by different dynamics. Finally, different emission pathways (e.g. $CO_2$, $N_2O$ and the resulting stratospheric water vapor feedback) will further change stratospheric background conditions (temperature, HOx,

NOx), with important consequences for stratospheric chemical reactions resulting in ozone production and depletion and thus, eventually, STE of ozone.

*(2) Lines 304-305: Can you be more specific on how " the slow-down hydrological cycle under SRM will further enhance this tropospheric humidity reduction"?*

We have deleted that sentence.

*Lastly for the geoengineering assessment to be practical, there needs to be a specific metric for measuring the impact. For example, the change in the distribution of tropospheric ozone in terms of the probability density distribution of surface ozone concentration might be useful.*

We agree that we need a specific metric for measuring the impact. We compared the geoengineering scheme with the reference case – RCP6.0. And we have done t-tests for each grid cell to understand whether the two scenarios are statistically different. In all maps, the hatched regions are insignificant, with $p > 0.05$.

*And the authors need to discuss what can be improved in the modeling effort in the discussion section.*

We have added sentences in Line 456-462:

"This study may be biased by the following factors: (1) using prescribed stratospheric aerosols does not allow the simulation of the full interactions between chemistry, aerosol microphysics, and dynamics. A fully interactive model including those interactions would be important. (2) The vertical resolution is not sufficient to produce an interactive QBO in the model used, which may also affect transport processes. (3) The model does not include the scattering effect of aerosols on tropospheric photolysis rates, which might lead to an overestimate of the UV enhancement in the troposphere."

**Reviewer #3**

*General comments:*
*This manuscript examined the effects of stratospheric sulfate aerosol and solar insolation reduction on tropospheric ozone and surface ozone. The study also examines the both chemical and transport mechanisms of tropospheric ozone changes to SRM techniques. The findings of this paper help us get a better understanding of effects of SRM. In general, I found the main points and the structure of this manuscripts are clear. Below are my comments for making the manuscript more concise. I recommended the paper to be published with minor revision.*

Thanks.

*Specific Comments:*

*Line 54: Add one or two references of sulfate aerosol effects in the stratosphere.*

"It is well known that sulfate aerosols in the stratosphere enhance heterogeneous chemical reactions that lead to enhanced ozone depletion after larger volcanic eruption (Solomon, 1999)."

*Caption for Figure S1: What do you mean about the 10N to 10S gridded present day MLS/OMI satellite data? Doesn't the data cover the extra-tropics?*

Zonal mean values from the satellite data were derived from a 10° x 10° gridded product. Model results were interpolated from the same grid.

*Line 159: move "the last 40 years of geoengineering" in line 161 to here.*

Changed.

*Line 163: Figure 3a and 3b: why there is significant temperature increase over north Atlantic? More explanations of temperature changes between SRM runs and RCP 6.0 would be helpful.*

We have added in Lines 178-180:

"The similar warming in the North Atlantic under G4SSA and G4SSA-S relative to RCP6.0 (Fig. 3a and 3b) is due to the regional cooling under RCP6.0 as a result of slowing down of the Gulf Stream (Hartmann et al., 2013)."

*Line 164-166: In Figure 3c: I did not see clear warming signal in Asia. And 'warming' here is confusing: both G4SSA and G4SSA-S show temperature decrease compared to RCP6.0. The red color in Figure 3c just means that G4SS4 has less temperature reduction than G4SS4-S.*

We have changed it to "The temperature difference between G4SSA and G4SSA-S (Fig. 3c) is larger in the Northern Hemisphere winter (Fig. S3)." (Lines 181-182)

*In Figure S3, surface temperature in G4SS4 does show a significant warming over northern Europe and Asia compared to RCP6.0 in winter. I think that is the feature that agrees with the characteristic "winter warming" from volcanic stratospheric aerosol (Robock, 2000).*

We have described this in Lines 182-184:

"The warming over northern Europe and Asia in G4SSA relative to G4SSA-S is the characteristic "winter warming" from volcanic stratospheric aerosol (Robock, 2000)."

*Line 168-170: around 50 hPa in the tropics, the G4SSA-S also shows the significant warming.*

The slight warming in the lower stratosphere might be a result of slight increases of ozone concentration there (Fig. 8b) and dynamic heating. We have added in Line 187-189:

"The slight warming in the lower stratosphere under G4SSA-S (Fig. 4b) is a result of ozone changes and dynamical heating (discussion in Session 3.3.2)."

*Line 173: would be better to switch the sequence of Figure 2b and Figure 2c*

Done

*Line 207: How the halogen changes over these three runs?*

We have added in Line 238-239:

"The halogen loading in the three scenarios is the same, and more information could be found in Morgenstern et al. (2017)."

The halogen loading is in Figure 1(b) of Morgenstern et al. (2017).

*Line 361: mid-high latitude: are you talking about the lower stratosphere      100 hPa? Adding the location of tropopause in Figure 8 would be much helpful.*

Yes.

Instead of tropopause, we have added the prescribed sulfate aerosol layer as a contour line in Figure 8 as suggested by reviewer #1 on page 5. This might be better to help understanding the stratospheric warming and ozone depletion.

*Line 364: temperature changes in which direction?*

We have added in Line 393-394:

"changes in stratospheric temperature (warming in G4SSA and cooling in G4SSA-S) also change the photochemistry of ozone."

*Line 364: "Altogether, this results in year-round lower stratospheric ozone loss world-wide that peaks during the return of sunlight at high SH latitudes." Which figure describes this feature?*

We have added Fig. S14

*Line 368: Figure 8b: Confused here: in Figure 4b, temperature shows an increase in the tropics around 50-70 hPa. Other regions show temperature reduction. You mentioned that ozone increase in Figure 8b is due to temperature decreases. While the regions with T increases (tropics, 50-70 hPa), ozone has a maximum increase.*

The slight warming in the lower stratosphere in Fig. 4b might be a result of slight increases of ozone concentration there (Fig. 8b). We have added in Lines 187-189:

"The slight warming in the lower stratosphere under G4SSA-S (Fig. 4b) is a result of ozone changes and dynamic heating (discussion in Section 3.3.2)."

We have added discussion in 3.3.2 line 399-404:

"However, in Fig. 4b, there is a slight warming around 50 mb in the tropics, where ozone concentration also shows a stronger increase (Fig. 8b). As tropospheric cooling results in a slow-down of the B-D circulation (Fig. 9b) (Lin and Fu, 2013; Nowack et al., 2015; Shepherd and McLandress, 2011), there is an increase of ozone in the tropical upwelling region, which leads to increasing temperatures there as ozone is a strong shortwave and longwave absorber."

*It would be much easier to understand if you mention lat/pressure when describing figures. Lower stratosphere in the polar region could reach as low as 400 hPa. Adding a tropopause in Figure 8 would be much helpful.*

We have added the latitude and the altitude in lines 390 and 392. We have added the injected sulfate aerosol as contour lines in Figure 8.

*Line 383-384: Why tropical upwelling response differently between this study and Aquila et al 2012? Please be more specific?*

We have added in Lines 416-419:

"The differences between previous studies and our result may be because some previous studies used fixed ozone, with different stratospheric heating rates. In addition, in previous studies, the QBO was interactively simulated and the models had a higher model top."

*Line 394: (Fig. S12). . delete one period*

Done.